# Ammonia and Humidity Sensing by Phthalocyanine–Corrole Complex Heterostructure Devices

**DOI:** 10.3390/s23156773

**Published:** 2023-07-28

**Authors:** Lorena Di Zazzo, Sujithkumar Ganesh Moorthy, Rita Meunier-Prest, Eric Lesniewska, Corrado Di Natale, Roberto Paolesse, Marcel Bouvet

**Affiliations:** 1Institut de Chimie Moléculaire de l’Université de Bourgogne, UMR CNRS 6302, Université de Bourgogne, 9 Avenue Alain Savary, 21000 Dijon, France; dzzlrn01@uniroma2.it (L.D.Z.); sujithkumar_ganesh-moorthy@etu.u-bourgogne.fr (S.G.M.); maria-rita.meunier-prest@u-bourgogne.fr (R.M.-P.); 2Department of Electronic Engineering, University of Rome Tor Vergata, Via Politecnico 1, 00133 Roma, Italy; 3Laboratoire Interdisciplinaire Carnot de Bourgogne, UMR CNRS 6303, Université de Bourgogne, 9 Avenue Alain Savary, 21000 Dijon, France; eric.lesniewska@u-bourgogne.fr; 4Department of Chemical Science and Technology, University of Rome Tor Vergata, Via della Ricerca Scientifica, 00133 Roma, Italy; roberto.paolesse@uniroma2.it

**Keywords:** corrole, phthalocyanine, molecular materials, organic heterojunction, gas sensors, conductometric transducers

## Abstract

The versatility of metal complexes of corroles has raised interest in the use of these molecules as elements of chemical sensors. The tuning of the macrocycle properties via synthetic modification of the different components of the corrole ring, such as functional groups, the molecular skeleton, and coordinated metal, allows for the creation of a vast library of corrole-based sensors. However, the scarce conductivity of most of the aggregates of corroles limits the development of simple conductometric sensors and requires the use of optical or mass transducers that are rather more cumbersome and less prone to be integrated into microelectronics systems. To compensate for the scarce conductivity, corroles are often used to functionalize the surface of conductive materials such as graphene oxide, carbon nanotubes, or conductive polymers. Alternatively, they can be incorporated into heterojunction devices where they are interfaced with a conductive material such as a phthalocyanine. Herewith, we introduce two heterostructure sensors combining lutetium bisphthalocyanine (LuPc_2_) with either 5,10,15-tris(pentafluorophenyl) corrolato Cu (**1**) or 5,10,15-tris(4-methoxyphenyl)corrolato Cu (**2**). The optical spectra show that after deposition, corroles maintain their original structure. The conductivity of the devices reveals an energy barrier for interfacial charge transport for **1**/LuPc_2_, which is a heterojunction device. On the contrary, only ohmic contacts are observed in the **2**/LuPc_2_ device. These different electrical properties, which result from the different electron-withdrawing or -donating substituents on corrole rings, are also manifested by the opposite response with respect to ammonia (NH_3_), with **1**/LuPc_2_ behaving as an n-type conductor and **2**/LuP_C2_ behaving as a p-type conductor. Both devices are capable of detecting NH_3_ down to 10 ppm at room temperature. Furthermore, the sensors show high sensitivity with respect to relative humidity (RH) but with a reversible and fast response in the range of 30–60% RH.

## 1. Introduction

Among gas-sensing materials, molecular materials have been extensively studied because they offer lots of possibilities to tune their electrical and optical properties and the intermolecular interactions that they can develop with the target species. Among molecular materials, porphyrinoids, namely phthalocyanines and porphyrins, have been highly studied [1,2,3,4]. They are characterized by large π-aromatic systems and can reversibly interact with many molecules, by H-bonds and dipole–dipole and van der Waals interactions, with their metal center offering coordination bonding. They offer the possibility of electron transfer with redox active species. This is the reason why they have been used in numerous applications, e.g., for air quality monitoring [5], for controlling the freshness of food [6,7], and also for the analysis of biomarkers in breath [8]. Another family of porphyrinoids has been introduced more recently in the field of chemosensors, namely corroles [9,10,11,12], which have been used as sensing materials associated with optical, acoustic, electrochemical, and conductometric transducers, as recently reviewed [13]. While porphyrins are characterized by an aromatic macrocyclic system containing 22 electrons, corroles are contracted porphyrins, with a molecular skeleton featuring corrin, the nucleus of vitamin B12. Corrole was reported for the first time in the early 1960s [14], but it has gained renewed attention recently after the discovery of simple synthetic routes for its preparation [15,16]. Considering conductometric sensors, because of the rather low conductivity of corroles, they are often associated with more conducting materials, e.g., carbon nanotubes [10] and reduced graphene oxide [17], both applied to the detection of nitrogen dioxide. Another way to use low-conducting materials in conductometric transducers is to incorporate them into heterojunctions. Thus, two types of molecular-material-based heterojunction devices have recently been reported, namely double-layer heterojunctions [18] and double lateral heterojunctions [19] (Figure 1). The latter were obtained by the depositing material via an electrodeposition technique, such as electropolymerization. Thus, starting from 2,3,5,6-tetrafluoroaniline, we deposited the perfluoropolyaniline [19], and from the zinc porphine, we deposited the corresponding polyporphine [20]. Very recently, we reported the first example of electrodeposited polycorrole, starting from 5,10,15-(4-aminophenyl)corrolato]copper(III) as the monomer [21]. In the case of double-layer heterojunctions, the deposition technique can be the evaporation under vacuum [22,23] or any solution processing technique [24]. The common point of these heterojunction devices is that the top layer is made of a more conducting material (Figure 1).

In the field of air quality monitoring, sensors have to be able to detect ammonia (NH_3_) with a limit of detection (LOD) below 25 ppm (mol/mol), which is the recommended exposure limit averaged over an eight-hour workday (source: NIOSH, USA). Heterostructures have already been reported for such an application, such as for reduced graphene oxide (rGO)-cobalt oxide [25] and rGO-polyaniline (PANI) [26] heterojunction devices, with a sensitivity of 1% at 50 and 20 ppm, respectively, but they operate in dry air or at only one RH value. Other hybrid inorganic–organic heterojunction have also been reported with a positive or negative response to NH_3_ [27].

In the present work, we report the use of Cu (III)-tris(pentafluorophenyl) corrole (CuTpFPC, **1**) and Cu (III)-tris(p-methoxyphenyl) corrole (Cu-(p-methoxy) TPC, **2**) as sublayers in molecular-material-based double-layer heterojunction devices (Figure 1a), combining them with a highly conducting molecular material, namely lutetium bisphthalocyanine, LuPc_2_. Due to its radical nature, LuPc_2_ exhibits high conductivity at room temperature and can be easily oxidized and reduced [28], which makes such a sensor highly sensitive to redox active species [2]. However, the transport properties of these heterojunction devices are determined by the nature of the charge carriers in the sublayer, which can be p-type, n-type, or ambipolar [29].

## 2. Experimental Section

### 2.1. Chemicals and Syntheses

Lutetium bisphthalocyanine (LuPc_2_) was synthesized according to the literature [30]. Dichloromethane was procured from a local supplier and was distilled before use in solution preparation. Tetrabutylammonium perchlorate (>98%) (TBAP) and copper acetate (Cu(OAc)_2_) were purchased from Sigma-Aldrich. Thin-layer chromatography (TLC) was performed on Sigma-Aldrich silica gel plates. Chromatographic purification of the reaction products was accomplished by using silica gel 60 (70–230 mesh, Sigma-Aldrich, St. Louis, MO, USA) as a stationary phase. Corrole free bases were synthesized according to the literature [31]. Copper complexes were obtained by adapting the reported method [32] to Cu (III)-tris (pentafluorophenyl) corrole (**1**) and Cu (III)-tris(p-methoxyphenyl) corrole (**2**) by the reaction of Cu(OAc)_2_ with tris (pentafluorophenyl) corrole and tris-(p-methoxy)phenylcorrole, respectively.

### 2.2. Cyclic Voltammetry

The electrochemical experiments were performed with a PGSTAT302N Autolab Metrohm potentiostat interfaced with Nova 2.1 software. Cyclic voltammetry was carried out on a three-electrode setup consisting of a glassy carbon disk (3 mm in diameter) as a working electrode, a platinum wire as a counter electrode, and Ag/AgCl (NaCl 3 M) as a reference, called Ag/AgCl hereafter, isolated from the solution by a salt bridge containing the same electrolyte solution as in the cell to prevent any leakage of NaCl into the cell (E_Ag/AgCl_ = −0.066 V vs. SCE). The working C disk electrode was soaked for 10 min in KOH (2 M), polished with 0.1 µm alumina, etched for 10 min in concentrated sulfuric acid (2 M), and sonicated for 10 min in water and then in absolute ethyl alcohol. The cyclic voltammograms were performed in CH_2_Cl_2_ containing 0.1 M tetrabutylammonium perchlorate, TBAP, as the supporting electrolyte. The solutions were deoxygenated for 10 min with argon, and positive overpressure of argon was maintained above the electrolyte during the entire measurement performed at room temperature.

### 2.3. Sample Preparation

ITO interdigitated electrodes (IDEs) deposited onto 1 × 1 cm^2^ float glass substrate and separated by 75 µm with 50 nm thickness were sonicated three times with CH_2_Cl_2_ and ethanol for 5 min at each step and dried in an oven for 1 h at 100 °C. The corroles were deposited on the ITO substrates with a new type of deposition method called the quasi-dip coating method that we developed (Figure 1), starting from 10^−4^ M CHCl_3_ solutions of **1** or **2**. This method is quite different from classical dip coating, since here the solution was poured into a Petri dish, which contains the target substrate, until the substrate was submerged in solution; then, the solution was sucked out slowly by slightly tilting the Petri dish up to 45° and dried at 100 °C. In this way, we obtained highly homogeneous surfaces for these materials compared to the classical solvent casting technique.

The corrole complex modified IDE was then transferred into a UNIVEX 250 thermal evaporator (Oerlikon, Bingen am Rhein, Germany), and LuPc_2_ was deposited as the top layer (50 nm in thickness as controlled by a quartz crystal microbalance) through classical thermal evaporation under secondary vacuum, at ca. 1.0 × 10^−6^ mbar, and a sublimation temperature of ca. 410–420 °C.

### 2.4. Spectroscopic Characterization of the Devices

The UV-Vis spectra of the heterojunctions coated on glass plates were recorded on a Varian’s Cary^®^ 50 spectrophotometer (Agilent, Santa Clara, CA, USA) using a Xenon flash lamp (Hamamatsu Photonics, Shizuoka, Japan) as an excitation source. The Raman spectra of the devices and corrole powders were acquired by using a Renishaw inVia Raman microscope using a 473 nm laser (Renishaw, Wharton, UK) as an excitation source.

### 2.5. Electrical and Gas-Sensing Measurements

The electrical and sensing measurements were performed according to the previously described method [31,32], always at room temperature (19–21 °C). The electrometer was controlled by self-made software via the GP-IB board. The current–voltage (I-V) characteristics were registered in the range of −10–+10 V, starting and finishing at 0 V bias [33]. Ammonia gas, at 985 and 98 ppm in synthetic air, and synthetic air were used from standard gas cylinders, purchased from Air Liquide, Paris, France.

The NH_3_-sensing experiments were performed dynamically through alternative exposure to different ammonia concentrations in the range of 10–90 ppm, at controlled relative humidity (RH). The required humidity in the chamber was produced through a humidity generator connected with the fluidic line and controlled in the range of 30–60% by a commercial humidity sensor (HMT-100, Vaisala, Vantaa, Finland). The system is semi-automated, in which the opening of the mass flow controller valves, mixing of the gases, control of the relative humidity (RH), and data acquisition were controlled by customized software.

## 3. Results and Discussion

### 3.1. Syntheses

The Cu complexes were obtained by synthetic methods adapted from the literature [31,32]. The electronic absorption spectra clearly show successful metalation (Appendix A). Thus, the Soret bands were modified, with a shift from 417 to 433 nm for copper complex **2**. The number of Q bands decreased from the metal free corroles to the copper complexes due to the change in symmetry.

### 3.2. Electrochemical Characterization

The cyclic voltammograms of corroles **1** and **2** (Figure 2) present two redox systems: one reduction step at $E_{1}^{\mathrm{Red}}=-0.13\;V\;\mathrm{vs}.\;\mathrm{Ag}/\mathrm{AgCl}$ and $E_{2}^{\mathrm{Red}}=-0.635\;V\;\mathrm{vs}.\mathrm{Ag}/\mathrm{AgCl}$, respectively, and one oxidation step at $E_{1}^{\mathrm{Ox}}=0.83\;V\;\mathrm{vs}.\;\mathrm{Ag}/\mathrm{AgCl}$ and $E_{2}^{\mathrm{Ox}}=0.285\;V\;\mathrm{vs}.\;\mathrm{Ag}/\mathrm{AgCl}$, respectively. It must be mentionned that substituents on the phenyl groups in the meso position have a strong influence on the potential position. Thus, when the three meso groups are (p-methoxy)phenyls, the potentials of the reduction and oxidation peaks are shifted by more than 500 mV in the negative direction compared to those obtained for corrole **1**, substituted by three pentafluorophenyl groups. These peaks are associated with the reduction and oxidation of the macrocyclic ring and can be used to estimate the energy values of the HOMOs and LUMOs’ frontier orbitals. Therefore, the onset values (Figure 2) versus SCE as the reference electrode can be reported using Equations (1) and (2) [34].
(1)$$E_{\mathrm{HOMO}}=-(E_{\mathrm{onset}}^{\mathrm{Ox}}+4.4)$$
(2)$$E_{\mathrm{LUMO}}=-(E_{\mathrm{onset}}^{\mathrm{Red}}+4.4)$$

The energies of the HOMO and LUMO of **1** are −4.90 and −4.28 eV, respectively, and −4.55 and −3.75 eV for **2**, respectively.

### 3.3. Device Characterization

The devices were obtained by successive deposition of a corrole complex by the solution processing technique described above, followed by vacuum evaporation of a 50 nm-thick LuPc_2_ layer.

#### 3.3.1. Spectroscopic Characterization

The optical absorption spectra of corrole complexes **1** and **2** deposited on glass with the quasi-dip coating technique highlight good deposition with good molecular dispersion on the substrate. The two different corroles show good adhesion and coverage on the plain glass. Compared to the CHCl_3_ solutions, the spectra are slightly broadened, with a red shift of the Soret and Q bands, by 7 nm for both bands of **1** at 410 and 560 nm, by 9 nm for the Soret band of **2** at 441 nm, and by 10 and 20 nm for its two Q bands, at 551 and 647 nm, which are actually shoulders (Figure 3). These red shifts indicate the formation of J aggregates associated with edge-to-edge intermolecular interactions in the solid state [35].

The bilayer devices **1**/LuPc_2_ and **2**/LuPc_2_ were obtained by coating films of **1** and **2** by thermal evaporation of the LuPc_2_ top layer. The optical absorption spectra confirmed that after deposition, the devices contain both materials (Table 1). Both heterostructures exhibit the same peak at 669 nm, which belongs to LuPc_2_ and corresponds to its Q-band (Figure 4) [36]. The bands at 605 and 604 nm correspond to the so-called “blue vibration band” of LuPc_2_. The peak at 460 nm that corresponds to a transition towards the semi-occupied molecular orbital of LuPc_2_ appears as a shoulder at ca. 460 nm in the spectrum of **1**/LuPc_2_ (Figure 4a) and is masked in **2**/LuPc_2_ by the Soret band of **2** (Figure 4b). Indeed, the main contribution of corrole complexes appears in the range of 350–500 nm (Soret bands) as broadened peaks, at 409 nm for **1** (Figure 4a) and at 439 nm for **2** (Figure 4b). Therefore, the spectra of the heterostructures appear as the superimposition of the spectra of both layers.

To achieve wider chemical characterization of the devices based on the corrole complexes and LuPc_2_, they were studied by Raman spectroscopy (Appendix A). Some peaks correspond to the sublayer, at 646, 1016, 1080, 1340 cm^−1^ (Cα-Cα bonds), and an intense peak at 1529 cm^−1^ attributed to C-F binding, while other peaks can be assigned to the top layer LuPc_2_, namely 578 cm^−1^ corresponding to Pc breathing, at 780 cm^−1^ to C=N aza breathing, at 1408 cm^−1^ to C_α_-C_meso_, and at 1601 cm^−1^ to C=C in the benzene ring [37,38,39]. Therefore, the Raman spectrum of the **1**/LuPc_2_ heterostructure shows peaks that can be attributed to the two layers (Figure 5).

The same feature appears for **2**/LuPc_2_, with some peaks that can be attributed to the sublayer at 661, 884.5, 982, 1075.5, 1196, 1292, 1312, and 1343 cm^−1^ (Cα-Cα bonds), while the main peaks of the LuPc_2_ top layer are well visible, namely at 680, 1122, and 1407 cm^−1^. Additionally, a few peaks are common to both materials, such as the peaks at 735 cm^−1^ (C-H wagging), 780 cm^−1^ (C=N aza stretching), 1177 cm^−1^ (C-H binding), 1221 cm^−1^ (C-H binding), and 1601 cm^−1^ (benzene stretching)_._ These results also confirm that the development of heterostructures with the two types of corrole complexes and LuPc_2_ is achieved correctly without decomposing the material.

#### 3.3.2. Morphological Characterization

The AFM images are very different for both bilayer devices. They showed that the fluorinated corrole **1**/LuPc_2_ device is very homogeneous, with very low roughness (RMS: 1.64 nm) (Figure 6 left below). The sample exhibits lots of short vertical needles (height: 2–3 nm). The **2**/LuPc_2_ device exhibits holes, ca. 10 nm in depth, with a few hills, only 6 nm in height (Figure 6 right below) and with very low roughness (RMS: 1.47 nm). However, the sample is not homogeneous, and a few big structures appear, up to 50 nm in height, with roughness up to 10 nm in some areas (Appendix A).

#### 3.3.3. Electrical Characterization

The electrical properties of the functionalized electrodes were investigated by recording the current–voltage (I-V) curves in the range from −10 to +10 V. The **1**/LuPc_2_ device exhibits non-linear I-V characteristics, showing the existence of an interfacial energy barrier between the two materials. The apparent energy barrier estimated from the tangent to the I-V curve at high bias was 1.15 V, which is a rather low value (Figure 7). On the contrary, the **2**/LuPc_2_ heterostructure exhibits linear behavior associated with ohmic contacts. The current values at 10 V of the former and the latter are of 6 × 10^−6^ A and 1.5 × 10^−5^ A, respectively. The I-V curves are symmetrical, as expected for such symmetrical devices [18].

### 3.4. Ammonia-Sensing Properties

The response of the devices toward ammonia was studied by submitting them to two types of exposure: long exposure (10 min) at a constant NH_3_ concentration (90 ppm) and short exposure (1 min) at different NH_3_ concentrations (10–90 ppm), with each exposure period being followed by recovery periods of 30 min and 4 min, respectively. For the **2**/LuPc_2_ device, the current decreases under ammonia and increases under clean air (Figure 8). Considering the electron-donating nature of ammonia, the device is of p-type, NH_3_ molecules neutralizing positive majority charge carriers, as for p-type resistors. The response and recovery times (t_90_) are ca. 6 min and 8 min, respectively, as evaluated from 10 min/30 min exposure/recovery cycles. The absolute response ΔI, defined as I_0_–I_f_, with I_0_ and I_f_ being the current values at the beginning and end of an exposure period, respectively, and consequently the relative response (RR), defined as 100 × ΔI/I_0,_ were calculated for the short exposure to NH_3_ (Figure 9). The absolute values of ΔI and RR increase almost linearly with the NH_3_ concentration up to 50 ppm; then, saturation of sensors occurs. From the linear part, the sensitivity S, defined as the slope of the RR = f([NH_3_]) curve, was 2.5% ppm^−1^.

For the **1**/LuPc_2_ heterojunction, long exposure to ammonia highlighted a sharp increase in current under NH_3_, as happens for n-type materials, indicating that the majority of charge carriers are electrons. The decrease in the majority positive charge carriers in LuPc_2_ induces a decrease in the energy barrier at the sublayer–top layer interface, and in turn, a current increase, as reported previously with n-type sublayers [23]. However, the sharp increase was followed by a slow decrease during exposure, and then, during recovery, a sharp decrease followed by a slow increase was observed (Figure 10 left). During short exposure/recovery cycles, the trends remained the same, i.e., sharp current increase and decrease were observed at the beginning of the new exposure and recovery periods, respectively (Appendix A). The response and recovery times are ca. 10 s, corresponding to the duration of the sharp increase, or decrease, in the current during the 10 min/30 min exposure/recovery cycles. However, as the exposure time is shorter, the current decrease during the total exposure period is negligible compared to the short time variations. More interestingly, the sharp current variations increase with the NH_3_ concentration, the absolute response (ΔI), and relative response (RR) increasing linearly between 10 and 70 ppm (Figure 9 right). The mean sensitivity value for the entire concentration range was ca. 0.6% ppm^−1^.

For a better understanding of this behavior, we submitted the device to a 500 mL min^−1^ flow of synthetic air at 60% RH for 2 h. Then, we performed the same short exposure/recovery cycles in the range of 10–90 ppm NH_3_, at 60% RH (Figure 10 right). It shows the same trend observed at 40% RH but with an increase in current which seems to be higher with the increase in RH. In that case, the absolute response (ΔI) and relative response (RR) increase linearly with the NH_3_ concentration in the entirety of the studied range (10–90 ppm) (Figure 9 right), and the sensitivity is 0.9% ppm^−1^. Clearly, the sensitivity increases with the RH value, even though the ΔI value is quasi unchanged in both experiments.

We confirmed this dependence of the response to NH_3_ on the RH value by performing experiments at a fixed NH_3_ value (50 ppm) in the range of 30–60% RH, with short exposure/recovery cycles (Figure 11). We have to point out the very good reversibility. Therefore, since the response to ammonia increases with the increase in the RH value, a humidity sensor should be coupled to the NH_3_ sensors, and a correcting factor could be calculated to take into account the humidity effect. Whatever the RH value, the current increases, indicating that the majority free charge carriers remain n-type, contrary to what we previously observed with ambipolar sublayers [22,40].

The difference in the nature of majority charge carriers between the two devices can be correlated with the frontier energy levels of the two corrole complexes. Due to the presence of pentafluorophenyl moieties, corrole **1** is easier to reduce and more difficult to oxidize, correlated to stabilization of its HOMO and LUMO orbitals, compared to Cu(p-methoxy)TPC, by ca. 0.5 eV, as depicted in the present electrochemical study. As a result, it is easier to inject electrons from the electrodes into the fluorinated corrole material.

### 3.5. Humidity-Sensing Properties

The response of the devices toward humidity was also studied by steps of 10 min, from 30% to 60% RH. This humidity range was chosen because, in most of the practical applications such as in industry or in air quality stations, gas sensors operate within this humidity range. For both devices, the current increases sharply with each 10% RH value increase and decreases sharply with each 10% RH decrease. The recovery is very good for **1**/LuPc_2_ (Figure 12) and the total for **2**/LuPc_2_ (Figure 13). The current value increases by 6.2% and 7.5% when the RH increases from 30 to 60% for **1**/LuPc_2_ and **2**/LuPc_2_, respectively. The former exhibits hysteresis, with the current remaining higher when the RH value decreases (Figure 12 right). The current variation as a function of the RH value is almost linear, with a linear fit leading to an R^2^ coefficient of 0.98 with the increasing and decreasing RH values. For the latter, only a small difference appears for the current between increasing and decreasing RH values (Figure 13, right).

## 4. Conclusions

Two types of heterostructures with two different corrole complexes, namely Cu(III)-tris (pentaFphenyl) corrole and Cu(III)-tris (p-methoxyphenyl) corrole, have been developed by combining them with an intrinsic molecular semiconductor, lutetium bisphthalocyanine. The devices have been characterized by optical measurements, such as UV-visible and Raman spectroscopies, and by AFM analyses. About the sensing properties, both devices showed different electrical properties and different behavior towards ammonia. The one with the Cu(III)-tris (p-methoxyphenyl) corrole complex exhibits linear I-V characteristics and a current decrease under ammonia, thus showing no heterojunction effect and its p-type behavior. In the case of the Cu(III)-tris (pentaFphenyl) corrole complex, the I-V characteristics are non-linear, and the current increases under ammonia, thus showing the n-type behavior of this heterojunction device, with electrons being easily injected from the electrode into the fluorinated corrole complex. This difference can be correlated with the frontier energy levels of the two corrole complexes. Due to the presence of pentafluorophenyl moieties, the Cu(pentaF)TPC is easier to reduce and more difficult to oxidize, correlated to stabilization of its HOMO and LUMO orbitals, compared to Cu(p-methoxy)TPC. Both devices are sensitive to NH_3_ below 10 ppm, so they fit the requirements for air quality monitoring. In addition, both materials show high sensitivity to humidity but with complete reversibility, which is rarely observed in conductometric sensors. Therefore, a humidity sensor should be coupled to the NH_3_ sensors, and a correcting factor could be calculated to take into account the humidity effect.

## Data Availability

Data are available near corresponding authors on demand.

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
