# Peer review of "Ammonia and Humidity Sensing by Phthalocyanine–Corrole Complex Heterostructure Devices"

_sensors, 2023, doi:10.3390/s23156773_

Round 1

Reviewer 1 Report

The authors fabricated NH3 and humidity sensors that were made of phthalocyanine-orrole complex heterojunctions. It is interesting. The following questions should be revised for improving the quality of the paper.

1. The SEM or TEM results should be included in the section of Results and discussion because the morphology is a primary factor to affect the gas-sensing properties of the sensors.

2. The standard deviation should be include in Fig. 8.

3. The response/recovery of the sensors based on phthalocyanine-orrole complex heterojunctions to the NH3 gas and humidity-sensing should be included in the section of Results and discussion.

4. The sensors herein responds to both ammonia gas and humidity. How to overcome the mutual interference of ammonia gas and humidity?

Author Response

Dear reviewer,

We thank you for your comments and suggestions that give us the opportunity to improve this manuscript. We answered point by point to all your comments.

The authors fabricated NH3 and humidity sensors that were made of phthalocyanine-orrole complex heterojunctions. It is interesting. The following questions should be revised for improving the quality of the paper.

  1. The SEM or TEM results should be included in the section of Results and discussion because the morphology is a primary factor to affect the gas-sensing properties of the sensors.

R: We carried out AFM analyses of the two devices. The AFM images are very different for both devices. They showed that the fluorinated corrole 1/LuPc2 device is very homogeneous, with a very low roughness (RMS: 1.64 nm) (Fig. 1 left below, Fig. 6 in the revised manuscript). The sample exhibits lots of short vertical needles (heigth: 2-3 nm).The 2/LuPc2 device exhibits holes, ca. 10 nm in depth, and with a few hills, only 6 nm in heigth (Fig. 1 right below), with a very low roughness (RMS: 1.47 nm). However, the sample is not homogeneous, and a few big structures appear, up to 50 nm in heigth, with a roughness up to 10 nm in some areas (Fig. S2 in the revised manuscript). We added these comments in the revised manuscript.

These AFM analyses were carried out thanks to the participation of Professor Eric Lesniewska. This is the reason why we added him in the list of authors.

Figure 1. AFM images of 1/LuPc2 (left) and 2/LuPc2 devices, from top to bottom: 2 mm x 2 mm 2D pictures and profiles corresponding to the 1 mm-long line shown on 2D pictures, respectively.

  1. The standard deviation should be include in Fig. 8.

R: We added the standard deviation for Delta I in Fig. 8, calculated from the successive cycles at each ammonia concentration.

  1. The response/recovery of the sensors based on phthalocyanine-corrole complex heterojunctions to the NH3 gas and humidity-sensing should be included in the section of Results and discussion.

R: We moved Figure S3 from SI to manuscript.

  1. The sensors herein responds to both ammonia gas and humidity. How to overcome the mutual interference of ammonia gas and humidity?

R: It is true that the sensors are highly sensitive to humidity variations, but we have to point out a very good reversibility. So, since the response to ammonia increases with the increase of RH value, a humidity sensor should be coupled to the NH3 sensors and a correcting factor could be calculated to take into account the humidity effect. This comment has been added in the revised manuscript.

Reviewer 2 Report

1.Abstract should contain, in nuce, the whole paper. It should emphasize the central idea and the key points it should also suggest any implications or applications of the research you discuss in the paper. But in this case, the abstract presents mostly general things. It should be revised, in my opinion.

2. The introduction should contains some literature review about the ammonia sensing ( the key point of this paper, however).

3. The ammonia sensor should be extensively  characterized: response time, recovery time.

4. The sensing mechanism should be explained in more detail.

5. The conclusions mut be reinforced.

Author Response

Dear reviewer,

We thank you for their comments and suggestions that give us the opportunity to improve this manuscript. We gave a response point by point to your comments.

1.Abstract should contain, in nuce, the whole paper. It should emphasize the central idea and the key points it should also sugg3est any implications or applications of the research you discuss in the paper. But in this case, the abstract presents mostly general things. It should be revised, in my opinion.

R: We thank the reviewer for his remark. We removed a few general sentences from the abstract and add a few more experimental results.

  1. The introduction should contains some literature review about the ammonia sensing ( the key point of this paper, however).

R: We added a paragraph in the introduction to talk about ammonia sensors.

  1. The ammonia sensor should be extensively  characterized: response time, recovery time.

R: For 2/LuPc2 device, the response and recovery times (t90) are ca. 6 min and 8 min , respectively, as evaluated from 10 min/30 min exposure/recovery cycles. For 1/LuPc2 device, the response and recovery times are ca. 10 s, corresponding to the duration of the sharp increase, or decrease, of the current during 10 min/30 min exposure/recovery cycles.

  1. The sensing mechanism should be explained in more detail.

R: Towards ammonia, the behavior of 2/LuPc2, which exhibits only ohmic contact, is the same as a p-type resistor. NH neutralizes positive charges carriers in the top layer, LuPc2, leading to a current decrease. For 1/LuPc2 that is an heterojunction device (with non-linear I-V characteristics), the decrease of majority positive charge carriers in LuPc2 induces a decreases in the energy barrier at the sublayer – top layer interface and in turns a current decrease, as reported by us, for example in M. Bouvet et al., Sensors Actuators B, Chemical, 2010, 145, 501-506 or in ref. 23 of the initial manuscript. These mechanisms have been added, in two different parts in section 3.4.

  1. The conclusions mut be reinforced.

R: We modified the conclusion. In particular, we mentioned the difference in current-voltage characteristics, as detailed in our response to reviewer 3.

Reviewer 3 Report

  Title: Ammonia and Humidity Sensing by Phthalocyanine-Corrole complex Heterojunction Devices The paper reports the ammonia and humidity sensing by Corrole-Pc complex heterojunction devices. First of all, the formation of a heterojunction device can't be confirmed from the optical absorption spectrum as the authors claimed in Fig. 3 and described on page-5 in the last paragraph. The absorption spectra in Fig. 3 just exhibit the presence of the materials LuPc/1 or LuPc/2 in the mixture.  As the heterojunction devices are made of LuPc/1 and LuPc/2, where in either case the bandgaps (HOMO-LUMO levels) of the materials are different so the current-voltage (I-V) characteristics should not be symmetric as shown in Fig. 6. The symmetric curve in Fig. 6 indicates no formation of the heterojunction device but is just as ohmic junction. If the heterojunction device was really formed then the I-V curve must be asymmetric. Therefore, the authors needs to recheck the formation of heterojunction devices, which is one of the main title of this paper. Once, the device structure is confirmed then the sensing properties could be validated.  

Author Response

Dear reviewer,

We thank you for you comments and suggestions that give us the opportunity to improve this manuscript. We gave a response point by point to your comments.

The paper reports the ammonia and humidity sensing by Corrole-Pc complex heterojunction devices. First of all, the formation of a heterojunction device can't be confirmed from the optical absorption spectrum as the authors claimed in Fig. 3 and described on page-5 in the last paragraph. The absorption spectra in Fig. 3 just exhibit the presence of the materials LuPc/1 or LuPc/2 in the mixture.  As the heterojunction devices are made of LuPc/1 and LuPc/2, where in either case the bandgaps (HOMO-LUMO levels) of the materials are different so the current-voltage (I-V) characteristics should not be symmetric as shown in Fig. 6. The symmetric curve in Fig. 6 indicates no formation of the heterojunction device but is just as ohmic junction. If the heterojunction device was really formed then the I-V curve must be asymmetric. Therefore, the authors needs to recheck the formation of heterojunction devices, which is one of the main title of this paper. Once, the device structure is confirmed then the sensing properties could be validated.  

R: We thank the reviewer for his comment. This is true, so, we replaced “The formation of heterojunctions was confirmed by measuring the optical absorption spectra” by ”The optical absorption spectra confirmed that after deposition the devices contain both materials”. Since heterojunction is related to particular electrical properties, we replaced “heterojunction device” by “heterostructure” in sections 3.3.1 and 3.3.2. Then, since 2/LuPc2 exhibits linear I-V curves, we replaced “heterojunction” by “heterostructure” all along the manuscript, keeping “heterojunction” for the 1/LuPc2 device. The behavior of 2/LuPc2 is comparable to this of CuPc/LuPc2 heterostructure, where both materials are p-type and the I-V curves are quasi linear, while 1/LuPc2 is comparable to this of Cu(F16Pc)/LuPc2 in which the n-type behavior of the sublayer is responsible for the positive response to ammonia (ref 18 in the initial version, V. Parra et al, Analyst, 2009).

We also replaced “heterojunction devices” by “heterostructure devices” in the title.

Round 2

Reviewer 1 Report

The authors have made the positive revisions for the questions so that the paper accept in the present form.

Reviewer 2 Report

No other suggestions. The paper was modified according to suggestions.